# Mandibular Radiolucencies: A Differential Diagnosis of a Rare Tumor

**DOI:** 10.3390/diagnostics12071651

**Published:** 2022-07-07

**Authors:** Antonio Cabrera-Arcas, Jose-Francisco Montes-Carmona, Luis-Miguel Gonzalez-Perez

**Affiliations:** 1Department of Oral and Maxillofacial Surgery, Virgen del Rocio University Hospital, 41013 Seville, Spain; aca92alm@gmail.com (A.C.-A.); josmoncar@gmail.com (J.-F.M.-C.); 2Department of Surgery, School of Medicine, University of Seville, 41009 Seville, Spain

**Keywords:** mandibular radiolucency, odontogenic tumors, Pindborg tumor, calcifying epithelial odontogenic tumor, differential diagnosis

## Abstract

The diagnosis and treatment of maxillofacial cystic lesions requires careful evaluation and correlation of the clinical presentation and radiological studies. The Pindborg tumor, also known as the calcifying epithelial odontogenic tumor, is a locally invasive benign neoplasm, with only around 300 cases being published to date. This study presents a new case of this already uncommon neoplasm, not associated with an impacted tooth, and describes the clinicopathological features of this rare entity, along with a review of other reported cases. Despite surgery having been recognized as the treatment of choice for the Pindborg tumor, no firm consensus exists concerning the extension of surgical resection.

## 1. Introduction

The calcifying epithelial odontogenic tumor was first described as a distinct odontogenic tumor by the oral pathologist Jens Pindborg in an abstract in 1955 and again in an article in 1958 [1,2]. The origin of this very rare benign tumor is controversial, and it is believed that it can be derived from the oral epithelium, reduced enamel epithelium, intermediate stratum, or even dental lamina remnants that are or have been a part of the human–tooth–formation process. The histopathological characteristics are composed of three elements: epithelium, amyloid plaques, and calcifications. This tumor usually has slow and asymptomatic growth, and there have been more than 300 cases published to date [3,4,5]. The objective of this article is to describe one additional case of this rare entity.

## 2. Case Report

A 52-year-old female was referred to our department for the evaluation of a facial pain located in the left preauricular area with two years of evolution. The main symptoms were impaired mouth opening, loss of more than 5% of her weight in the last 6 months, and intense continuous pain in the affected region with extension to the left ear that led to the diagnosis of temporo–mandibular disorder. Clinical history relating to the condition showed no response to conservative treatment with medication, physiotherapy, and occlusal splint. The left lower first and second molars were extracted in a dental clinic one year ago. The patient’s medical history included arterial hypertension and degenerative arthropathy, with shoulder and neck involvement. She had no history of maxillofacial trauma, and she denied a history of smoking, alcohol, or the use of any other drugs.

A physical examination revealed discrete swelling in the left mandibular side, and movement was severely limited, with a maximum jaw opening of 18 mm. Facial nerve function was intact, no symptoms indicating the involvement of the inferior alveolar nerve were detected, and there was no evidence of cervical lymphadenopathy. During an intraoral examination, a hard nodule in the left retromolar trigone was observed. The remainder of the head and neck exploration was normal.

The usual diagnostic imaging study with a panoramic radiograph showed a well-defined mixed radiolucent–radiopaque lesion with peripheral enhancement in the left mandibular ramus occupying almost its entire thickness (Figure 1). Computed tomography performed with contrast was used to confirm the existence of a heterogeneously enhancing 3 cm long tumor with lobulated margins and bicortical expansion without bone perforation. Radiographically, there appeared to be an expansile cystic mass with areas of calcification and sclerotic borders. The mandibular lower border was intact (Figure 2A,B).

A preoperative study was carried out that did not contraindicate surgical intervention. Under general nasotracheal anesthesia, an intraoral vestibular incision was used to approach the tumor resection. An initially intraoperative biopsy was taken of the lesion, which was diagnosed as suggestive of a myxoid lesion. During surgery, the lesion proved to be primarily cystic in nature and could be removed with marginal resection from the surrounding mandible with preservation of bone continuity. No other action was deemed necessary to ensure bone stability. The surgically excised lesion was 5 × 4 cm in size. The specimen was examined by the Department of Pathology and revealed islands of polyhedric epithelial cells with nuclear pleomorphism, prominent intercellular bridges, and eosinophilic acellular plaques resembling amyloid and Liesegang rings calcifications. Immunohistochemical analysis revealed strong positivity for cytokeratin CK5/6 and p63. The resection margins appeared clear of tumor presence, and on that basis, the authors concluded that the tumoral lesion had been successfully removed. The final diagnosis of a primary intraosseous Pindborg tumor was established (Figure 3).

There were no postoperative complications. The patient was discharged on the fourth postoperative day and followed up at the outpatient clinic. A pureed diet was advised for the first month and a normal diet thereafter. Clinical and radiological imaging studies showed excellent long-term stability of the mandible. The maximum jaw opening was stable at 40 mm with no evidence of mandible deviation. There was no evidence of recurrence after a follow-up of one and a half years.

## 3. Discussion

Odontogenic tumors represent between 2% and 5% of all the neoplasms of the maxillofacial area. In the fourth edition of the World Health Organization Classification of Head and Neck Tumors, twenty-four odontogenic tumors were listed, and the Pindborg tumor, also known as the calcifying epithelial odontogenic tumor, remains a benign odontogenic subtype without significant modifications and is considered a rare pathological entity representing less than 1% of all odontogenic tumors [6].

The clinical behavior is variable and usually presents as an intraosseous or central lesion, as in this case report. Extraosseous or peripheral lesions account for less than 5% of cases. The literature reports that the Pindborg tumor usually starts as painless intumescences that cause a slow bone expansion. It can show up at any age, but is more common in middle age, with a peak incidence in the fifth decade of life, and there is no sex predilection [6]. It usually develops in the posterior region of the mandible, as in our patient, with a mandibular to maxillary proportion of 3:1, and may present clinical and pathological similarities to other odontogenic lesions. It has been proposed that the Pindborg tumor stems from the stratum intermedium of the dental organ or from the dental lamina. The radiological aspects observed are variable and depend on the time of evolution of lesions and can present in different phases: only radiolucent lesions, radiolucent–radiopaque images as in our case, or only dense radiopaque areas. Pindborg tumors are mostly unilocular, but they can also be multilocular. Evidence of bone destruction is often present, and mottled densities caused by calcifications and ossification, described as a “driven snow appearance”, can occasionally be seen. In some cases, particularly in tumors of a relatively short period, the calcification is very small and may not be defined on radiographs. Nearly 50% of cases are associated with an unerupted tooth, which could not be proven in our patient because of a previous history of exodontia. When an unerupted tooth is incorporated with the tumor, the radiopacity is usually located near the tooth crown [3,4,7].

A review of the current scientific literature revealed some similarities between our case and those of previous authors. In our patient, a diagnosis was not made preoperatively. A malignant tumor was suspected, given the clinical characteristics of the lesion (intense pain that did not respond to treatment, unexplained weight loss, and imaging diagnostic study). Misinterpretation of benign and malignant odontogenic tumors has been documented, and malignant transformation has also been reported, although it is exceptionally rare and could be a source of diagnostic confusion that suggests an erroneous diagnosis of a metastatic carcinoma of unknown origin [8,9,10]. In diagnostic imaging studies, no pathognomonic findings are associated with these tumors. In our case report, the radiological differential diagnosis included a dentigerous cyst, an adenomatoid odontogenic tumor, and a calcifying odontogenic cyst [3,4,5,11,12]. According to the literature, the Pindborg tumor exhibits considerable variation in its histologic appearance, and several entities may also be considered in the differential diagnosis. On the basis of clinicopathological characteristics, the following diagnoses can usually be excluded: odontoma, ameloblastomas (including those previously known as ameloblastic fibroodontoma), giant cell granuloma, fibroma, myxoma, and cemento-ossifying fibroma.

The surgical treatment of odontogenic tumors continues to be a subject of discussion. Simple conservative enucleation and curettage have a high rate of recurrence and can be followed by persistence or recurrence of the tumor, which frequently leads to the need for later surgical excision. Radical resection including mandibular margins, using either marginal or segmental resection, is also controversial, although there appears to be a high success rate and it could be justified for aggressive lesions, as in our case, where the preservation of surrounding anatomical structures, including the dental nerve and basal mandibular cortex, was taken into account for rehabilitation in a second stage with an implant-supported prosthesis to improve the patient’s mandibular function [13]. Targeted therapies could be of benefit in treating these neoplasms as a substitute for surgical resection or as possible neoadjuvant therapy. Because of the rarity of this entity, clinical trials can be difficult to perform without multicenter collaboration [14,15,16].

## 4. Conclusions

Mandibular radiolucencies are very common and are difficult to diagnose. When considering the surgical risks involved in the removal of the lesion, the differential diagnoses of benign and malignant tumors are of great importance. The Pindborg tumor is a benign, locally invasive odontogenic tumor, not always associated with an impacted tooth, with a high recurrence rate. In extremely rare cases, it may show aggressive growth and signs of malignancy. Lesions should be treated as soon as possible because these neoplasms usually develop and invade nearby structures. The definitive diagnosis of this case report aims to reinforce the importance of accurate radiographic analysis as well as clinical correlation to arrive at a diagnosis and determine the optimal treatment planning of apparently aggressive lesions of the mandible. Surgical options include appropriate marginal resections with tumor-free margins in order to reduce the probability of local recurrence. After surgery, patients need periodic clinical and radiographic follow-ups.

## Figures and Tables

**Figure 1 diagnostics-12-01651-f001:**
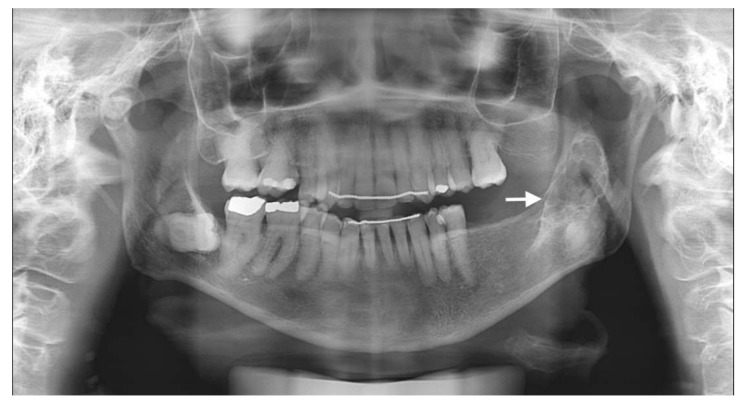
Preoperative panoramic radiographic image. Radiolucency with many radiopaque spots described as driven snow appearance. No association with impacted teeth was seen (see arrow).

**Figure 2 diagnostics-12-01651-f002:**
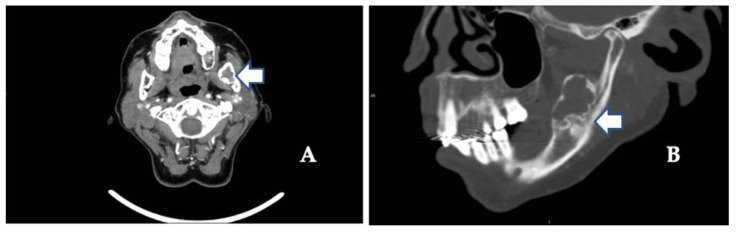
(**A**) Contrast-enhanced computed tomography scan showing the extent of the tumor along the mandibular angle (see arrow) (coronal image). (**B**) Sagittal view shows mixed radiolucent–radiopaque lesions (see arrows) and advanced osteoarthritic changes in the left temporomandibular joint (see arrow).

**Figure 3 diagnostics-12-01651-f003:**
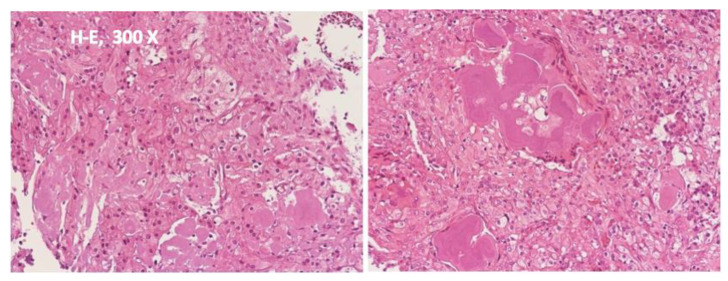
Examination of the surgical specimen showed polygonal epithelial cells with prominent intercellular bridges associated with amyloid and the presence of eosinophilic concretions forming annular structures (Liesegang’s rings) and calcifications. The margins of the surgical resection appeared to be tumor-free. No further treatment was recommended at this stage. Histopathological diagnosis indicated a calcifying epithelial odontogenic tumor (Pindborg tumor).

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
