# Peer review of "Mandibular Radiolucencies: A Differential Diagnosis of a Rare Tumor"

_diagnostics, 2022, doi:10.3390/diagnostics12071651_

Round 1

Reviewer 1 Report

Dear Authors,

The article 'Mandibular radiolucencies: A differential diagnosis of a rare tumour' was to present a new case of this already uncommon neoplasm, not associated with an impacted tooth, and describes the clinicopathological features of this rare entity, along with reviewing other reported cases.

Minor spell check required. American English is required.

Punctuation mistakes should be corrected. 

Article is prepared careless. Authors did not use MDPI guidelines. Loss of introduction and clinical case description.

Case description is presented as figures description. It is mistake. 

Necessary to separate the case report as an article section.

Prepare references using MDPI guidelines

The article is well planned and prepared. It contains a decent summary of the analyzed topic. 

Article must be rejected. Reconsider after major revision.

Author Response

Review Report (Reviewer 1): 

We greatly appreciate your thoughtful comments that helped to improve our manuscript. Following your recommendations, we rewrote our manuscript to reflect more clearly our findings. Correction of grammatical errors and English improvement were carried done as suggested. Regarding the reviewer's comments, we added Introduction section and clinical case description as Case report section. Thank you for nothing the absence of this statement.

Regarding your valuable comments, in accordance with these explanations, we have altered the Introduction, Case report, Discussion, and References sections accordingly.

Reviewer 2 Report

Review on “Mandibular radiolucencies: A differential diagnosis of a rare tumour

General comment

The authors describe through 3 images the case of a Pindborgtumor. As this tumor is extremely rare, only case reports or very small case series have been published so far. Thus, I believe this further case could enrich the literature as it well describes the clinical and radiological presentation and histopathological diagnosis of this rare disease. The images are of good quality, and the article is generally well written. However, the English language could be further improved. I hereby attach few specific comments. 

1) Figure 1:

When describing the patient’s history, it is not clear from the text whether the patient suffered from any disease in general.I believe one sentence should be spent to describe the patient’s comorbidities. Did she suffer from any chronic disease (e.g., diabetes mellitus, hearth diseases, rheumatic diseases, etc.)? Was she a smoker?

2) Figure 2: 

Did the patient have any symptoms indicating possible involvement of the inferior alveolar nerve, considering the location of the tumor near the mandibular canal? As the CT scan was performed with contrast medium, I believe it would be interesting to describe the behavior of the tumor in regard to the contrast (e.g., was there any enhancement?). I believe it would be interesting, if possible, to add an intraoperative image of the surgical resection that was used to remove the tumor between figure 2 and figure 3.

3) Figure 3:

After the resection of the tumor, did the authors place a titanium plate in order to increase mandibular stability and decrease the risk of fracture? Looking at the CT scan in figure 2, the healthy inferior border of the mandible seems rather thin. Thus, considering the type of surgical resection and the CT scan images, I wonder what was the thickness of the mandible that the authors were able to preserve, and whether they put any plate to reduce the risk of fracture. Did the Pathologist perform any immunohistochemistry? If so, what were the findings?

Author Response

Review Report (Reviewer 2):

We highly appreciate the reviewer helpful comments on our manuscript. Thank you for your kind and encouraging words that have enabled us to improve our paper. More specifically, we answer the following questions:

Case report section:

The most important tools in the diagnosis and treatment of tumoral lesions are, without doubt, careful clinical examinations and detailed reports on the patient’s history and symptoms. Due to your useful suggestions, in the revised manuscript the authors have included in lines 38-40 and 43-44 a reference to the patient´s medical history. Thank you for nothing the absence of this statement.

Regarding your additional comments, the entire paragraph between lines 49-54, 69-72 and 75-78 about diagnostic imaging study, bone stability and immunohistochemical analysis was modified and we have inserted a specific explanation in the Case-report and Discussion sections.

Discussion section:

Regarding the reviewer's comment on involvement and preservation of anatomical structures, we have also included in lines 138-143 some of our criteria on the treatment followed in this case. Thank you for nothing this.

 References section:

To explain this, we have also included a new citation in the References section (Infante-Cossio P, Prats-Golczer V, Gonzalez-Perez LM et al. Experimental and Therapeutic Medicine. 2013; 6: 579-583).

Regarding your valuable comments, in accordance with these explanations, we have altered the Introduction, Case report, Discussion, and References sections accordingly.

Round 2

Reviewer 1 Report

Article can be accepted after Editor decision.